**Data Availability Statement:** Since it's a protocol, we don't have data at the momment. The data will

# Evaluation of the community-based outpatient therapeutic feeding program implementation for managing children with severe acute malnutrition in Northwest Ethiopia: A mixed-method evaluation protocol

**Wubet Worku Takele**[1]*, **Amare Demsie Ayele**[2], **Tsegaye Gebremedhin Haile**[3], **Ayal Debie**[3], **Ashenafi Tazebew Amare**[4], **Chalie Tadie Tsehay**[3], **Eskedar Getie Mekonnen**[5]

1 Department of Community Health Nursing, School of Nursing, College of Medicine and Health Sciences, University of Gondar, Gondar, Ethiopia, 2 Department of Pediatrics, Child Health Nursing, and Public Health Nutrition College of Medicine and Health Sciences, University of Gondar, Gondar, Ethiopia, 3 Department of Health Systems and Policy, Institute of Public Health, College of Medicine and Health Sciences, University of Gondar, Gondar, Ethiopia, 4 Department of Child Health and Pediatrics, School of Medicine, College of Medicine and Health Sciences, University of Gondar, Gondar, Ethiopia, 5 Department of Reproductive, Institute of Public Health, College of Medicine and Health Sciences, University of Gondar, Gondar, Ethiopia

* wubetakele380@gmail.com

## Abstract

### Background

Community-based outpatient therapeutic feeding program (C-OTP) in Ethiopia has been launched to manage uncomplicated severe acute malnutrition (SAM) by trained Health Extension Workers (HEWs). This program is believed to be the most effective strategy for reaching a large group of children suffering from SAM in rural and disadvantaged communities. Nonetheless, poor treatment outcomes, notably mortality and prolonged recovery time, become pressing public health problems, which could be a result of suboptimal implementation and poor service quality.

### Objectives

To evaluate the implementation of C-OTP for managing uncomplicated severe acute malnutrition in the Central Gondar Zone.

### Methods

Multiple studies involving both qualitative and quantitative will be conducted. Availability of essential drugs and equipment, acceptability of the program by mothers/caregivers, health extension workers' compliance to the treatment protocol, and treatment outcome will be assessed employing different methods. Likewise, knowledge of health extension workers about SAM diagnosis and management and their skills to diagnose and manage uncomplicated malnutrition will be determined. Health extension workers, mothers/caregivers, supervisors, and healthcare administrators will be enrolled in the study. Besides, children's

be publicized once gathered and available at our hand.

**Funding:** This work was supported by the International Institute for Primary Health care-Ethiopia (IPHC-E). The funders had and will not have a role in study design, data collection and analysis, decision to publish, or preparation of the manuscript.

**Competing interests:** The authors have declared that no competing interests exist.

medical records registered between 2017 and 2020 will be reviewed to determine the treatment outcome. The data will be collected using pretested self-administered and face-to-face interviewer-administered questionnaires. Similarly, focus group discussions (FGDs), in-depth interviews, and observation checklists will be applied. Binary logistic regression analysis will be conducted, while the qualitative data will be analyzed using thematic content analysis.

## Discussion

Severe acute malnutrition is a public health problem that remains the underlying cause for over half of under-five mortality in Ethiopia. As a result, community-based therapeutic care has been launched in the country to address these problems and maximize population-level impact by improving treatment coverage, access, and cost-effectiveness. Despite its achievement, the program has been threatened with unfavourable treatment outcomes and a shortfall of resources. Hence, this implementation evaluation study will also identify gaps between healthcare systems and service users. The output will help programmers pass evidence-based and sound decisions to tackle the key barriers.

## Introduction

Globally, malnutrition results in 3.1 million child deaths annually, accounting for 45% of all under-five mortality [1]. Severe acute malnutrition (SAM), in particular, is the most devastating and life-threatening form of malnutrition, with a 20–30% case fatality rate. The prevalence of wasting in Ethiopia is 7% [2]. In cognizance of the extent of the problem, Ethiopia has launched a community-based outpatient therapeutic program (C-OTP) to manage uncomplicated SAM by trained HEWs. The program was formulated with core operating principles such as maximum population coverage and access, timeliness, appropriate care, and care for as long as needed [3]. The community-based approach is home-based management of SAM that enables timely detection of malnutrition and is provided for those without medical complications using ready-to-use therapeutic feeding (RUTF) or other homemade nutrient-dense diets [4]. The program has multiple indicators from input to impact that could be evaluated using various evaluation dimensions and model indicators.

Nearly three-quarters of the children diagnosed with SAM are uncomplicated (the majority have a good appetite for RUTF/diets and do not have co-morbidities) and can be managed in the outpatient program. Despite this, the caseload is still concentrated in secondary and tertiary level health facilities, making it challenging to provide appropriate service [5].

The traditional in-patient care model of therapeutic feeding centres (TFC) could not effectively respond to large-scale humanitarian crises, as poor access was the prime obstacle [6]. Parallel to this, the in-patient management approach could be the underlying cause for delay in receiving treatment which could further lead to multiple long and short-term complications, including but not limited to heart failure and marked immunosuppression resulting in infectious diseases [1, 7]. It is, therefore, vital to deliver C-OTP that is integrated with a facility-based approach and implemented on a large scale with competent HEWs [1, 8].

There is controversial evidence on the effectiveness of the C-OTP [9–11]. On the one hand, C-OTP has declared a more feasible approach in that the mean cost per treated child was reduced by more than half (134.8$ versus 284.6$) [10, 12]. On the other hand, the death rate in

C-OTP was 1.2%, while it is zero in the TFC [10]. Likewise, 4 out of 10 children with SAM remained severely malnourished at discharge after undergoing the recommended management and length of stay in outpatient management [8, 13]. Further, a report in Ethiopia highlighted that the recovery rate among children with SAM admitted to C-OTP was 45%, with an average weight gain of 5–6 g/kg/day, which is far below the recommended weight gain from the program [11]. However, the community-based approach provides care at an affordable cost where the cost-effectiveness of emergency community-based therapeutic programs varies from US$12 to US$132 per year of life gained [3].

Despite the sizable impact of the C-OTP on reducing cost, addressing a large group of the child population, decreasing the chance of contracting illness due to extended hospital stay, and having an enabling environment to implement it, some evidence indicated the possible limitations [1, 8, 14]. More significant than expected poor treatment outcomes and longer recovery time in this model of care than in inpatient care might be related to its poor implementation.

Therefore, this study will evaluate the implementation of the C-OTP in the central Gondar zone. Assessing the program's performance, key programmatic variables, and clinical outcomes is pivotal in bringing strategies to end childhood death and associated long-term sequelae due to SAM.

## Evaluation questions

1. To what extent are HEWs knowledgeable about SAM diagnosis and management principles and capable of diagnosing and managing uncomplicated SAM?

2. Do the HEWs adhere to the C-OTP management guideline?

3. Do mothers/caregivers of children with SAM accept the C-OTP?

4. What are the barriers and enablers for the observed level of C-OTP implementation?

## Objectives

- To assess the skills of HEWs on case identification and management of uncomplicated SAM.

- To determine the knowledge of HEWs about SAM.

- To measure the compliance of HEWs to the C-OTP protocol.

- To explore the mother's/caregiver's acceptability of C-OTP of the service.

- To explore the barriers and enablers to the observed level of integrated community-based outpatient therapeutic program implementation for SAM.

- To determine the treatment outcome of severely malnourished children who received treatment under the C-OTP.

## Evaluation methods

### Evaluation design and settings

Multiple case studies with mixed-method evaluation will be used to evaluate the C-OTP implementation from the 15th of September to the 30th of December 2021 in the Central Gondar Zone, Amhara Region. The zone is situated 727 km away from Addis Ababa, the country's

capital. There are 11 districts and 880 HEWs in the region. A minimum of two and a maximum of three HEWs are deployed in each health post.

Amhara regional state is an impoverished region among all areas in the country that is threatened by the high burden of food insecurity and malnutrition. Multiple strategies, including C-OTP, have been applied to address malnutrition and prevent its associated morbid and fatal complications. The program launched nationally in 2007, with HEWs to support and manage children with SAM having no medical complications by administering medicines and therapeutic food. HEWs should conduct periodic community screening and manage cases based on severity. Severely malnourished children without medical complications are eligible to enrol in the program. They receive RUTF and antibiotics (regardless of clinical presentations). During each visit, HEWs assess patient's progress weekly and follow them until they fulfil the discharge criteria. HEWs counsel mothers/caregivers about feeding and other healthy behavior practices to optimize the treatment outcome.

According to the treatment guideline, HEWs' are required to refer children with medical complications to the nearby stabilization center to receive in-patient care, as they may need parenteral medications, nasogastric feedings, and close follow-up. Once children are free from medical complications, clinicians discharge them from in-patient care and link them to the C-OTP.

## Study participants and reporting procedure

All children aged 6–59 months with uncomplicated SAM enrolled in C-OTP, mothers/caregivers, HEWs, program officers, and health extension supervisors will be involved. The OTP follow-up records and reports available in each health post will be considered.

The finding of the study will be reported using different reporting guidelines for each study design/approach employed. Accordingly, the Strengthening the Reporting of Observational Studies in Epidemiology (STROBE) and COREQ (Consolidated Criteria for reporting qualitative research) writing checklists for the cross-sectional and qualitative analyses, respectively, will be used [15, 16].

## Sample size and sampling procedures

A total of three districts will be included. The number of participants recruited will be determined by each objective of the study. All HEWs available in the three selected districts will be included to assess their knowledge and diagnosis skill. Similarly, mothers/caregivers whose children completed the treatment within the last six months will be invited and enrolled to determine the service's acceptability. All HEWs in the three districts and health posts will also be included in assessing the compliance and availability dimensions. Further, all children enrolled in the treatment centers between 2017 and 2020 will be included to determine the treatment outcome (death, transferred out, failure to respond, default, and cured/survived). Medical recording charts with incomplete records for outcome variables will be excluded.

A case study and descriptive qualitative study will be conducted. All health posts in the three districts will be considered as cases to identify the program's facility-related barriers and enablers. Moreover, the purposive sampling technique will be applied to approach the study participants. The participants will be recruited considering different criteria, including their experience in the given position, active engagement in the program, and caseload (participants from the lowest and highest case flows). In-depth interviews with HEWs, key informants' interviews with program managers (nutrition officers and HEWs supervisors) at the district level, and focus group discussions (FGDs) with mothers/caregivers will be conducted until the level of data saturation reaches. The data saturation will be determined during data collection

and analysis. The interviewers will stop collecting data when new information is no longer generated from the interviewee [17]. In addition, the data will be transcribed and analyzed parallelly with the data collection, and data collectors will quit when new themes/codes are not produced.

## Evaluation dimensions and variables

Availability and acceptability dimensions from the access framework and the compliance dimension from the implementation fidelity framework will be used to evaluate the program implementation [18–20]. The dependent variables will be compliance to the protocol, HEW's knowledge, case identification and management capacity of the HEWs, SAM treatment outcome (transferred out, default, death, and cured/recovered), mothers/caregivers program acceptability, and barriers as well as enablers of the program. A list of independent variables will be selected per the individual to community-level dimensions. These variables are the sociodemographic, economic, and cultural characteristics of mothers/caregivers.

**Availability.** The physical availability and functionality of essential equipment needed for the program implementation [20]. The data will be gathered using resource inventory checklists comprising of 14-items (Table 1). If all essential equipment required to provide the care like Plumpy'nut, medications, Mid Upper Arm Circumference (MUAC) tape, and wall charts, are available and functioning well, it will be stated as 'fully available' otherwise 'not fully available.'

**Table 1. Summary of the dimensions/variables of implementation evaluation of C-OTP and measurements in Central Gondar zone, Ethiopia, 2021.**

| Framework | Dimensions/variables | Measurement items | Data collection technique |
|---|---|---|---|
| Access [18, 20] | Availability | • Checking for the treatment protocol<br>• Checking for essential drugs and medicines<br>• Checking for anthropometry measuring devices | • Observation checklists |
| | Acceptability | • Approach with HEW<br>• The RUTF safety to children<br>• Distance from the health post<br>• Transportation access | • Face-to-face interview |
| Implementation fidelity [23] | HEW's knowledge of SAM | • Diagnosis of SAM<br>• Management of SAM | • Self-administered questionnaire |
| | HEWs diagnostic and treatment capacity | • Diagnosis of SAM<br>• Management of SAM | • Observational |
| | Compliance/adherence of HEWs with the treatment protocol | • Does the HEW document all the findings?<br>• Does the HEW give medicines and RUTF as per the guideline?<br>• Does the HEW record the anthropometry measurements?<br>• Does the HEW admit children based on the protocol? | • Chart review |
| Treatment outcomes | Died | • Evidence recorded on the OTP card | • Chart review |
| | Recovered | • Evidence recorded on the OTP card | • Chart review |
| | Defaulter | • Evidence recorded on the OTP card | • Chart review |
| | Transferred out | • Evidence recorded on the OTP card | • Chart review |
| | Dropout | • Evidence recorded on the OTP card | • Chart review |

**Acceptability.** Mothers'/caregivers' acceptance (satisfaction) about the service provided [20]. It will be measured using 14-item questions with five-point Likert scale responses ranging from 'strongly agree' to 'strongly disagree' (Table 1). The individual scores will be summed up. Acceptability will be defined as 'good acceptance' if a participant responds to ≥75% of the total number of questions (53) 'agree/strongly agree' (for positively coded questions) and 'disagree/strongly disagree' (for negatively coded questions); otherwise, 'poor acceptance.'

**Compliance/Adherence.** The adherence of HEWs to the treatment guideline will be measured by patient-provider interaction observations and document reviews using the 11-item checklists. The questions will assess both the diagnosis and treatment adherence of the HEWs. The responses will be categorized as 'good compliance' if all questions are addressed and 'poor adherence' if one of the eleven questions is missed.

## SAM treatment outcomes

- **Transferred-out**: if the child's condition has deteriorated and referred to a stabilization center for better treatment [21].

- **Cured**: a child fulfils the discharge criteria, namely MUAC≥11.5cm, no oedema, and weight-for-height >-2SD [21].

- **Defaulter**: a child absent for two consecutive visits [21].

- **Died**: a child dies when receiving treatment in the C-OTP [21].

- **Failed to respond**: a child doesn't reach the discharge criteria for SAM after 16 weeks of treatment [21].

**Uncomplicated SAM.** Children 6–59 months old whose weight-for-height/length below 70% or below -3z score of the World Health Organization (WHO) standards and bilateral pitting oedema and/or MUAC <11.5cm (age older than six months) and have no other medical complications like pneumonia and anemia [21, 22].

**Diagnosis and management skill.** The HEWs' will be deemed as 'skilful' in diagnosing and managing uncomplicated SAM if they assess, classify, and manage the child appropriately per the treatment guideline. An observational checklist will be used to evaluate the performance of each HEW. Their skills will be judged and determined by trained general practitioners/experienced paediatrics nurse specialists.

**Knowledge about uncomplicated SAM and management.** A pretested structured questionnaire will be administered, and then HEW's knowledge score will be categorized as per the estimate of the principal component analysis (PCA).

## Data collection tools, procedures, and quality assurance

Multiple data collection techniques and tools will be used. A face-to-face interviewer-based validated questionnaire will be used to assess the mothers/caregivers' acceptability of the service. Mothers/caregivers whose children have received the care each health post in the previous six months of the survey will be tracked by the respective HEWs and included in the study. A self-administered questionnaire (to assess the knowledge of HEWs), structured observation (for adherence, diagnosis skill, and availability), and document/chart review (for treatment outcome) checklists will be employed. Similarly, a semi-structured interview guide for the in-depth interview, key informants' interview, and focus group discussion (FGD) will prepared and used. Further, field notes during observation will be taken. Independent tools comprising

various indicators to assess the availability and supply of essential equipment for the program and HEWs compliance with the treatment guideline will also be used.

An observational checklist will be employed to assess the HEW's case identification and management skill. Accordingly, trained general practitioners will verify the accuracy of the HEWs' anthropometrics measurement, diagnosis, treatment performance, and treatment plan. Moreover, the knowledge of HEWs about SAM will be examined using a pretested self-administered questionnaire comprising the diagnostic methods, treatment principles, etc. Kobo Toolbox data collection software will be used to create the online data collection template and gather the data except for the self-administered questionnaire. The link will be sent to each data collector via email, WhatsApp, and telegram address. The participants will be briefed about the techniques to collect and submit the data properly. The principal investigator will check the delivery of the data and its completeness by signing in using the user account, and necessary actions will be undertaken. The passcode for the Kobo Toolbox account will not be shared with others who are not involved in the study as co-author.

The qualitative data will be triangulated, incorporating observational, FGDs, in-depth interviews, and key informant interviews. Data gained from field notes and interviews will be used to support the data obtained through the observation checklist. Using a flexible and piloted interview guide, the discussion and interviews will be underway by experienced interviewers. In other words, senior PhD students studying public health whose first language is Amharic, the local language, and who have experience as FGD moderators and individual interviewers will be recruited. The interview will be recorded using tape recorders.

All tools will be developed in English, translated to Amharic, and then back to English to ensure consistency. The face and content validity of the questionnaire will be done by a panel of reviewers who are nutritionists, paediatrics and child health specialists, and monitoring and evaluation experts. A pilot test will be conducted on the non-selected health posts. A reliability test will be done, and a Cronbach alpha of $\geq 0.7$ will be considered as the tool is reliable. Four nurses and two general practitioners/pediatric nurses (MSc holders) will be recruited for data collection and supervision, respectively. A two-day training about data collection procedures and participant handling will be delivered for data collectors and supervisors. Inter-rater variability among observers will be managed by referencing each item in the checklist and employing data collectors with similar work experience and educational status. Stringent supervision by trained supervisors will be done, and daily joint discussions with supervisors and the research team will be undertaken to identify gaps and give feedback on the data collection process.

Experts will transcribe the qualitative data after repeatedly listening to the audio recordings. Moreover, the data collectors will introduce themselves, the aim of the study in detail (not to evaluate/judge them) and establish a rapport with the participants to minimize the Hawthorne effect during observation [24, 25].

## Data management and analysis

**Quantitative data.** A nutritionist under the guidenace of a statistician will do the data management and analysis. The statistician will assist the nutritionist in the measurement of variables and their interpretation. A data dictionary document will be prepared. Data on acceptability, compliance, adherence, treatment outcome, and the skills of HEWs about management and diagnosis of SAM collected by the Kobo toolbox online data collection platform will be downloaded in an excel spreadsheet. Likewise, data on knowledge of HEWs about management and diagnosis of SAM will be entered into EpiData version 4.6 software and exported to an excel spreadsheet. The data will be merged and exported to STATA version 14 for further

analysis. Data cleaning will be completed, and the procedure followed with actions taken will be recorded in an excel spreadsheet. Missing data will be managed using proper imputation methods/models based on the nature of the variables (i.e., categorical or continuous). Descriptive statistics will be computed, and the results will be presented using tables, graphs, and narrations. A binary logistic regression will be employed for outcomes like HEW's knowledge, skills, and acceptability. The basic binary logistic regression assumption (e.g., chi-square) will be checked. The participant's age (in years), work experience (in years), training experience, level of education, and income will be used as independent variables that will be fitted in the multivariable logistic regression model to examine the association with HEWs knowledge and diagnosis skills. Similarly, waiting time, travel distance, mother's/caregiver's age, income, marital status, educational status, and employment status will be tested for association with the mother's/caregiver's acceptance. Continuous variables such as age and year of experience will not be categorized to build in the multivariable model [26, 27]. Considering the few numbers of independent variables in this study and to better control the confounding effect, the 'full-model fit' technique will be used [28]; all independent variables will be transferred to the multivariable logistic regression regardless of using a cutoff value such as p-value to transfer from the bivariable to multivariable analysis. The association of all independent variables with each outcome variable will be examined. Variables having independent associations with the outcome variables mentioned above will be identified based on the Adjusted Odds Ratio (AOR) and p-value corresponding to 95% CI. Variables with a p-value of $\leq 0.05$ will be considered statistically significant.

**Qualitative data.** The recorded audio will be transcribed, and the script to be translated to English will be developed. The data will be entered into NVivo version 12 software. The six-phase thematic data analysis procedure will be applied [29]. The researchers will repeatedly read the script and identify the meaningful units to create codes, collate codes into themes, review themes, label the generated themes, and report the findings. Before creating the themes and subthemes, disagreements between the coders will be resolved. A codebook will be generated accordingly.

A thematic analysis approach will be made. Themes and subthemes will be presented based on the findings and the existing literature in similar studies. Besides, a word-association test will be applied to identify the outranked concepts and phrases reflected by participants, and the results will also be compared using the chi-square test [30].

**Integration of the qualitative and quantitative data.** A concurrent mixed-method approach will be applied; the quantitative and qualitative data will be collected parallelly. Before starting the qualitative data collection, researchers will critically evaluate the quantitative tool to extract discussion points and concepts uncovered during the quantitative data collection to raise and incorporate in the interview guide. The data integration will be applicable for acceptability, availability, and compliance. The mothers' acceptability determined using a quantitative approach will be supported by data from mothers'/caregivers' interviews. The quantitative data on the availability and functionality of the essential equipment in each health post will also be backed by qualitative data gained through field notes and interviews. Similarly, the compliance determined by the quantitative approach will be supported by interviews of HEWs and observations during reviewing children's recordings.

## Ethical considerations

The study protocol has been submitted and approved by the Ethical Review Board (IRB) of the University of Gondar. Reference number: V/P/RCS/05/836//2021, issued on 23/022021. The study's progress report will be submitted to the International Institute for Primary Health

Care-Ethiopia (IPHC-E) coordinating office. Since the study will not involve invasive procedures, oral informed consent will be obtained from mothers/caregivers, HEWs, health extension supervisors, and other critical informants. Before the data collection, participants will be briefed about the aim of the research and their right to withdraw from the study at any time if they feel uncomfortable to continue. The study participants will receive compensation in monetary means (for the qualitative only) that will cover their travel and other related expenses related to the time they spend in the data collection. Mis-classified/diagnosed children by HEWs will be treated, and severe cases exhibiting medical complications will be referred to the nearby stabilization centres.

Children's names will be anonymized during chart review before the data collectors begin data collection, and identification numbers will be given to each chart. The collected data will be kept secure in the locked cupboard, and the data will be shared on a reasonable request.

## Discussion

Severe acute malnutrition is a significant public health problem contributing to over half of the under-five mortality in Ethiopia [31]. The C-OTP has been launched to reduce mortality and maximize population-level impact by improving treatment coverage, access, and cost-effectiveness [3]. The program was founded on three essential premises: First, if malnourished children gain access to nutritional care early in the evolution of their condition, the success rates may be high. Second, mothers/caregivers must understand, accept, and participate in the programs to attain the program goal and intended outcomes. Lastly, for the programs to move toward sustainability, there must be an up-front investment in social mobilization to ensure that critical stakeholders can benefit from the positive feedback from consumers [3].

Despite the substantial achievements of the C-OTP in reaching a large number of children with SAM, the program has been threatened with untoward treatment outcomes (death and dropout), lack of community engagement, shortfall of resources, scarce of competent HEWs, and misuse of resources [6, 32]. Likewise, the program's effectiveness and efficiency have also been an area of disagreement.

This implementation evaluation research will identify gaps related to healthcare systems and service users to bring comprehensive findings that reflect the program's different dimensions that have contributed to the effectiveness of the C-OTP in Ethiopia. First, the study will reveal the HEW's competency in managing uncomplicated SAM. Then healthcare-related factors, including a shortfall of necessary materials for community-based management of uncomplicated SAM, will be identified. Moreover, the study will show the program's acceptability by the beneficiaries. To this end, program managers, policymakers, and other stakeholders will utilize the finding to enhance the program's performance and pass evidence-based and sound decisions to track the program.

### Amendments of the study

Any acceptable change made during the data collection period, participant recruitment, methods of analyses, and other addition and deletion of methods will be declared during reporting manuscripts.

### Supporting information

**S1 Checklist.**
(DOCX)

## Acknowledgments

We would like to acknowledge data collectors, supervisors, and funders.

## Author Contributions

**Conceptualization:** Wubet Worku Takele, Eskedar Getie Mekonnen.

**Data curation:** Wubet Worku Takele, Amare Demsie Ayele, Ashenafi Tazebew Amare, Eskedar Getie Mekonnen.

**Formal analysis:** Wubet Worku Takele, Amare Demsie Ayele, Tsegaye Gebremedhin Haile, Ashenafi Tazebew Amare, Chalie Tadie Tsehay, Eskedar Getie Mekonnen.

**Funding acquisition:** Wubet Worku Takele, Amare Demsie Ayele.

**Investigation:** Wubet Worku Takele, Ayal Debie, Ashenafi Tazebew Amare, Eskedar Getie Mekonnen.

**Methodology:** Wubet Worku Takele, Tsegaye Gebremedhin Haile, Chalie Tadie Tsehay, Eskedar Getie Mekonnen.

**Project administration:** Wubet Worku Takele, Amare Demsie Ayele, Ayal Debie, Ashenafi Tazebew Amare, Chalie Tadie Tsehay, Eskedar Getie Mekonnen.

**Resources:** Wubet Worku Takele, Tsegaye Gebremedhin Haile, Ayal Debie, Chalie Tadie Tsehay, Eskedar Getie Mekonnen.

**Software:** Tsegaye Gebremedhin Haile.

**Supervision:** Wubet Worku Takele, Amare Demsie Ayele, Tsegaye Gebremedhin Haile, Ayal Debie, Ashenafi Tazebew Amare, Chalie Tadie Tsehay, Eskedar Getie Mekonnen.

**Validation:** Wubet Worku Takele, Tsegaye Gebremedhin Haile, Ayal Debie, Eskedar Getie Mekonnen.

**Visualization:** Wubet Worku Takele, Ashenafi Tazebew Amare, Chalie Tadie Tsehay, Eskedar Getie Mekonnen.

**Writing – original draft:** Wubet Worku Takele, Amare Demsie Ayele, Tsegaye Gebremedhin Haile, Ayal Debie, Eskedar Getie Mekonnen.

**Writing – review & editing:** Wubet Worku Takele, Amare Demsie Ayele, Tsegaye Gebremedhin Haile, Ayal Debie, Ashenafi Tazebew Amare, Chalie Tadie Tsehay, Eskedar Getie Mekonnen.

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
