## [Decision Letter · Decision Letter 0]

8 Apr 2022

PONE-D-21-25442

Evaluation of the community-based outpatient therapeutic program implementation for management of children with severe acute malnutrition: a mixed-method evaluation protocol.

PLOS ONE

Dear Dr. Takele,

Thank you for submitting your manuscript to PLOS ONE. After careful consideration, we feel that it has merit but does not fully meet PLOS ONE’s publication criteria as it currently stands. Therefore, we invite you to submit a revised version of the manuscript that addresses the points raised during the review process.

I would like to sincerely apologize for the delay you have incurred with your submission. It has been exceptionally difficult to secure reviewers to evaluate your study. We have now received two completed reviews; the comments are available below. The reviewers have raised significant scientific concerns about the study that need to be addressed in a revision.

Please revise the manuscript to address all the reviewer's comments in a point-by-point response in order to ensure it is meeting the journal's publication criteria. Please note that the revised manuscript will need to undergo further review, we thus cannot at this point anticipate the outcome of the evaluation process.

We look forward to receiving your revised manuscript.

Kind regards,

Miquel Vall-llosera Camps

Senior Editor

PLOS ONE

Journal Requirements:

2. In the Methods section please provide additional information regarding the qualitative study methodology. In particular, please describe how participants will be recruited for the study, whether an interview guide will be use, and the training and background of the interviewers.

Furthermore, please provide additional information regarding the questionnaire development and validation. Please also include a copy of the questionnaire as supporting information.

Reviewers' comments:

Reviewer's Responses to Questions

**Comments to the Author**

1. Does the manuscript provide a valid rationale for the proposed study, with clearly identified and justified research questions?

Reviewer #1: Yes

Reviewer #2: Partly

2. Is the protocol technically sound and planned in a manner that will lead to a meaningful outcome and allow testing the stated hypotheses?

Reviewer #1: Partly

Reviewer #2: Partly

3. Is the methodology feasible and described in sufficient detail to allow the work to be replicable?

Reviewer #1: No

Reviewer #2: No

4. Have the authors described where all data underlying the findings will be made available when the study is complete?

Reviewer #1: No

Reviewer #2: No

5. Is the manuscript presented in an intelligible fashion and written in standard English?

Reviewer #1: No

Reviewer #2: No

6. Review Comments to the Author

You may also provide optional suggestions and comments to authors that they might find helpful in planning their study.

Reviewer #1: This is a study protocol for an implementation evaluation of the critically important public health concern that severe acute malnutrition presents among children when they are most vulnerable to poor nutrition. The rationale presented by the authors is clear and the research question and proposed methodology is justified.

More detail is required on the proposed methods to ensure sufficient detail to allow replication/reproduction. For example, what approach will be taken for thematic analysis (inductive, deductive, etc.) or describe what type of thematic analysis (reflexive, codebook, etc.) An implementation evaluation framework is described but there is no reference to relevant sources that support the approach proposed.

There is no mention to appropriate reporting guidelines to ensure the rigorous reporting of results.

While there is mention by the authors of the data being available on completion of the study it is not clear that this will be in an open source repository?

Finally, the paper includes spelling and grammatical errors that should be attended to in order to ensure the article is clear and unambiguous.

Thank you for the opportunity to review this manuscript. Consideration of the points raised will, I believe, lead to an improved report and a thorough account of your intended research.

Reviewer #2: The protocols attempts to evaluate the implementation of Community-based Outpatient Therapeutic Program (C-OTP) for managing uncomplicated severe acute malnutrition in Central Gondar Zone using mixed-method evaluation. Though the manuscript address an interesting topic of the study it is not suitable for publication in its current form, since there is unclear or incomplete scientific reasoning on methodology presented and data analysis in this manuscript. Following are some of the concerns to address, before taking a decision.

1. My primary concern is that the manuscript mentions that the study data collection completes by 30th of December 2021, and after the completion of data collection, publication of protocol limits the purpose of the publication.

2. The manuscript failed to document the triangulation/integration of both quantitative and qualitative approaches. Moreover, the methods section of this manuscript is vaguely presented and discussed about the proposed data collection, management, role and responsibilities of its team and analytical methodology which needs clear explanation to understand study and implementation. For example, more description on research design, study area, assumptions for sample size calculation, sampling framework, inclusion/exclusion criteria, analytical framework/analysis of data, timeframe, and data safety and sharing mechanisms. Also the manuscript has not followed any standard guidelines and lacks clear definition and measurement of outcome specific indicators and their components/ validation of tools/scales used. Though here and there, the author(s) stated about all these but these needs to be structured in data and methods section.

3. Definition and measurement of outcome variables needs further more discussion. In other words, though the protocol proposes multiple composite indexes (viz., for Availability, Acceptability, Compliance and indicators of knowledge about uncomplicated SAM) to address the study objective with different scales, the authors failed to describe the components of each of the indexes and its validation. The study requires involvement of a statistician and the data analysis plans vaguely discussed in this protocol.

4. The write up of the protocol is too clumsy. Most of the time readability compromised and requires structural and careful review with professional language editing. The manuscript ends rather abruptly.

In conclusion, though the novelty of this protocol is limited, the experimental design, methods, expected outcomes may beyond state of art. However, before taking a decision for possible consideration of publication, the manuscript needs further more description and revision on its structure, completeness and language.

7. PLOS authors have the option to publish the peer review history of their article (what does this mean?). If published, this will include your full peer review and any attached files.

Reviewer #1: **Yes: **Dr Anne Griffin

Reviewer #2: No

---

## [Author Response · Author response to Decision Letter 0]

10 May 2022

Point-by-point response 

Dear editor and reviewers, thank you for giving us critical comments. The comments have a paramount positive impact on improving the readability and acceptability of the manuscript. We have provided responses to each concern and question. Kindly find it here below. 

Editor’s comments:

1. In the Methods section please provide additional information regarding the qualitative study methodology. In particular, please describe how participants will be recruited for the study, whether an interview guide will be used, and the training and background of the interviewers. Furthermore, please provide additional information regarding the questionnaire development and validation. Please also include a copy of the questionnaire as supporting information.

Response: Comment accepted, and necessary revisions have been made. A copy of the questionnaire has been attached. Moreover, the validity and reliability procedures are now well presented in the revised version of the manuscript.

2. We suggest you thoroughly copyedit your manuscript for language usage, spelling, and grammar. If you do not know anyone who can help you do this, you may wish to consider employing a professional, scientific editing service. 

Response: the comment has been accepted. A native academic speaker did the proofreading.

3. In your Data Availability statement, you have not specified where the minimal data set underlying the results described in your manuscript can be found. 

Responses: Commented accepted. Although this is a protocol that doesn’t obviously include results, we have described the data availability plan.

Reviewer #1

1. More detail is required on the proposed methods to ensure sufficient detail to allow replication/reproduction. For example, what approach will be taken for thematic analysis (inductive, deductive, etc.) or describe what type of thematic analysis (reflexive, codebook, etc.).

Response: Commented accepted.

2. An implementation evaluation framework is described but there is no reference to relevant sources that support the approach proposed. There is no mention to appropriate reporting guidelines to ensure the rigorous reporting of results.

Response: Dear, we have cited the source from which we have taken the frameworks. You are right; there should be reporting guidelines for each study research approach employed. We have accepted the comments. See the updated version on page…

3. While there is mention by the authors of the data being available on completion of the study it is not clear that this will be in an open-source repository?

Response: The data availability plan has been described. We will supply the dataset as supplementary material while submitting the manuscripts for publication.

Reviewer #2:

1. My primary concern is that the manuscript mentions that the study data collection completes by the 30th of December 2021, and after the completion of data collection, the publication of the protocol limits the purpose of the publication.

Response: Dear, we genuinely share your concern, and we believe that the issue should be raised. Nevertheless, the protocol was submitted back in August, and the review process is unfortunately delayed. Considering the nature of the manuscript (protocol), we had contacted the journal several times to expedite the reviewing process; unfortunately, it wasn’t possible. However, although now the data collection has been finalized, the actual data collection had been delayed because of the COVID and local security issues, and we believe that publishing the protocol still is worthwhile. Dear, we believe that publishing the protocol would provide knowledge to the prospective scholars to replicate and follow whenever they want to carry out similar studies; it will serve as a guide. We trust that it’s rare to see the protocols of evaluation studies published. Thus, this will be served as a guide. In fact, the full-length articles that will be produced and published from the project may serve as a guide, however, the objectives of the project will be split and published separately. Thus, comprehensive information for future researchers who want to evaluate the same program may not be available. All in all, although the study has already been completed, the protocol will help prospective scholars in planning the same study in different corners of the world. 

2. The manuscript failed to document the triangulation/integration of quantitative and qualitative approaches. 

Response: Comment accepted, and detailed descriptions of the procedures are now presented in a separate subheading.

3. Moreover, the methods section of this manuscript is vaguely presented and discussed the proposed data collection, management, role and responsibilities of its team, and analytical methodology which needs a clear explanation to understand the study and implementation. For example, more description of research design, study area, assumptions for sample size calculation, sampling framework, inclusion/exclusion criteria, analytical framework/analysis of data, timeframe, and data safety and sharing mechanisms.

Response: Dear, exact and relevant information has been provided. Comment accepted, and we substantially revised the section considering the untouched areas. Of the responsibilities of each team member, we believe it’s uncommon to see such narrations in such protocols, and we didn’t find this in any reporting guidelines available in the EQUATOR network (https://www.equator-network.org/). Of course, during a systematic review, it’s common to see in protocols who will do what. However, if you believe it is necessary, we are ready to modify it in the next revision. 

Dear, we didn’t calculate the sample size, as there are few participants/cases (health posts) in the three districts. We planned to consider all health posts and HEWs in the districts. Similarly, we didn’t use any random sampling technique and didn’t present a sampling frame; we will employ participants consecutively. We could have incorporated more than three districts and used a random sampling technique; however, there are limited resources (time and money). The details are presented under ‘sample size and sampling procedure’ on page #4. Dear, we don’t have particular criteria to exclude participants other than the criterion we have already stated for exclusion criteria. The data analysis, study design, and area are precisely presented. Moreover, the ethical consideration section explains the data safety and sharing mechanisms. 

4. Also, the manuscript has not followed any standard guidelines and lacks a clear definition and measurement of outcome-specific indicators and their components/ validation of tools/scales used.

Response: Dear, as you know, there is no unique guideline for evaluation studies to prepare and report the protocol, unlike systematic review and trial studies. However, as the approaches we will follow are qualitative and quantitative studies, we have followed guidelines developed for observational (cross-sectional) and qualitative studies Consolidated Criteria for reporting qualitative research (COREQ). The manuscripts will be reported according to the respective recommended guidelines. We have added the procedures used to frame the protocol in the newly revised manuscript. 

Sources: 

https://cdn-links.lww.com/permalink/acadmed/a/acadmed_89_9_2014_05_22_obrien_1301196_sdc1.pdf

https://academic.oup.com/intqhc/article/19/6/349/1791966

https://www.equator-network.org/wp-content/uploads/2015/10/STROBE_checklist_v4_cross-sectional.pdf

Note: we didn’t supply those reporting guidelines as there are no separate guidelines for protocols, as noted earlier. 

5. The definition and measurement of outcome variables need further discussion. In other words, though the protocol proposes multiple composite indexes (viz., for Availability, Acceptability, Compliance, and indicators of knowledge about uncomplicated SAM) to address the study objective with different scales, the authors failed to describe the components of each of the indexes and its validation. 

Responses: Comment accepted, and necessary modifications have been made. We have provided very brief information regarding the validation in the data quality section. We didn’t give each outcome headings not to repeat texts and make the reading tedious. As the validation and reliability assuring procedures are common to all variables, we don’t believe that a detailed description of all outcomes is required. 

6. The write-up of the protocol is too clumsy. Most of the time, readability is compromised and requires structural and careful review with professional language editing. The manuscript ends rather abruptly.

Response: An academic native speaker has done proofreading.

Thank you!

Wubet Worku Takele(t)he corresponding author

---

## [Decision Letter · Decision Letter 1]

22 Jun 2022

PONE-D-21-25442R1Evaluation of the community-based outpatient therapeutic program implementation for managing children with severe acute malnutrition in Northwest Ethiopia: a mixed-method evaluation protocol.PLOS ONE

Dear Dr. Takele,

Thank you for submitting your manuscript to PLOS ONE. After careful consideration, we feel that it has merit but does not fully meet PLOS ONE’s publication criteria as it currently stands. Therefore, we invite you to submit a revised version of the manuscript that addresses the points raised during the review process.

The reviewers have raised a number of concerns that need attention. They request additional information on methodological aspects of the study and detailed revisions to the statistical analyses being planned.

We look forward to receiving your revised manuscript.

Kind regards,

Thomas Phillips, PhD

Staff Editor

PLOS ONE

Reviewers' comments:

Reviewer's Responses to Questions

**Comments to the Author**

1. Does the manuscript provide a valid rationale for the proposed study, with clearly identified and justified research questions?

Reviewer #1: Yes

Reviewer #2: Yes

2. Is the protocol technically sound and planned in a manner that will lead to a meaningful outcome and allow testing the stated hypotheses?

Reviewer #1: Partly

Reviewer #2: No

3. Is the methodology feasible and described in sufficient detail to allow the work to be replicable?

Reviewer #1: No

Reviewer #2: Yes

4. Have the authors described where all data underlying the findings will be made available when the study is complete?

Reviewer #1: No

Reviewer #2: Yes

5. Is the manuscript presented in an intelligible fashion and written in standard English?

Reviewer #1: No

Reviewer #2: No

6. Review Comments to the Author

You may also provide optional suggestions and comments to authors that they might find helpful in planning their study.

Reviewer #1: Thank you to the authors for taking careful consideration of the previous review of this manuscript. It reads much more clearly and is more descriptive of the proposed research.

However, there are some points that require further detail:

Methods:

The context/description of the C-OTP programme remains vague. I suggest the inclusion of a diagram or similar illustration to describe the care pathways.

There is frequent reference to the 'protocol' - I am assuming that this is a protocol for the delivery of the C-OTP. However, it could also be the SAM guide (reference 19). Could a clear reference be provided and the protocol made available as supplementary material?

There is some explanation as to who is carrying out the data collection provided at the end of the methods section. More clarity as to their exact role would be appreciated. For example, lines 204-205 - what are the 'departments' that are referred to (clinical/university/government?).

Suggest the authors reconsider or define the use of data saturation as an indicator that all available information has been gathered using qualitative methods (lines 122).

The access framework proposed is from 1974 (line 125). Given the recent significant development of implementation frameworks, why have the authors chosen this?

Line 133 - the definition of 'availability' is confined to equipment/resources required to support the C-OTP. This seems at odds to the research aim and programme provision. It is also not the definition held within the cited access framework reference.

A reference is required for the Hawthorne effect (line 217).

What framework/theory will support the deductive analysis of the qualitative data? The authors mention 'existing literature' - has an evidence synthesis of existing literature been performed? (line 234-235).

What methodology does the 6-step process of qualitative analysis and coding refer to? (I'm assuming Braun & Clarke?).

Line 240: I am unclear whether the "questions and concerns" to discuss at interview arise from previously completed surveys/questionnaires? A timeline to support the flow process of the evaluation study would help to clarify.

Edits/Grammar:

This has much improved but there are some unresolved grammatical errors. There are acronyms used throughout that are not fully explained at first use, e.g., MUAC. There is also inconsistency in the application of C-OTP (lines 6, 27 and 45). This should be double-checked throughout the manuscript. I would also suggest that appropriate language is considered in the discussion, line 273, and the use of "victims" to describe the population of interest in this research.

A remarkable amount of work is being considered within this protocol and I appreciate the authors efforts to date to consider that it will provide future guidance to similar research.

Reviewer #2: I would like to congratulate the author(s) to adequately addressing reviewers concerns to some extent satisfactorily. While the authors have made some amendments to the protocol which improves the text, the concerns I have in the definition of outcomes measures and proposed methods of analysis of data (analytical framework) remain and failed to address the concerns raised in the first round of review process. The authors suggested to (a) strengthen methods section of this protocol and needs clear explanation to understand project. The protocol has not discussed role and involvement of a statistician and the data analysis plans also vaguely discussed in this protocol (instead of ‘binary Binary logistic regression analysis will be conducted’, discuss/list out the type, definition and measurement of select outcomes/dependent variables along with possible independent variables). (b) discuss the data collection and management process, data entry (electornic/double data data entry, data quality validation and assurance; (c) since the data collection completed before the publication of the protocol, it is suggested to present the date of completion of data collection and final status of data along with any inclusions/exclusions, missing values etc., (d) discuss the filed level operation issues and lessons learnt; possible measures adopted etc., during the study data collection/implementation.

In conclusion, the subject addressed in this study protocol is worth of investigation and recommend for possible consideration.

7. PLOS authors have the option to publish the peer review history of their article (what does this mean?). If published, this will include your full peer review and any attached files.

Reviewer #1: **Yes: **Dr Anne Griffin

Reviewer #2: **Yes: **Ramesh Poluru

---

## [Author Response · Author response to Decision Letter 1]

30 Jun 2022

Date: 29/06/2022

Point-by-point response 

Dear editor and reviewers, thank you for giving us critical comments. The comments have a major positive impact on improving the readability and acceptability of the manuscript. We have responded to each concern and question. Kindly find it here below. 

Reviewer #1

1. The context/description of the C-OTP program remains vague. I suggest the inclusion of a diagram or similar illustration to describe the care pathways.

Response: Dear, your comment is found to be valid. We have added detailed descriptions of how the program is being operated. 

2. There is frequent reference to the 'protocol' - I am assuming this is a protocol for delivering the C-OTP. However, it could also be the SAM guide (reference 19). Could an apparent reference be provided and the protocol made available as supplementary material?

Response: Yes, you are correct that ‘protocol’ refers to the SAM guideline. We use the term treatment protocol/guidelines interchangeably. We have revised the manuscript. The country uses the national SAM guideline for both in-patient and outpatient SAM management. There is no unique guideline for C-OTP. 

3. There is some explanation as to who is carrying out the data collection provided at the end of the methods section. More clarity as to their exact role would be appreciated. For example, lines 204-205 - what are the 'departments' that are referred to (clinical/university/government?).

Response: comment accepted. 

4. Suggest the authors reconsider or define the use of data saturation as an indicator that all available information has been gathered using qualitative methods (lines 122).

Response: comment accepted. The definition of data saturation has been provided in the revised version of the manuscript. 

5. The access framework proposed is from 1974 (line 125). Given the recent significant development of implementation frameworks, why have the authors chosen this?

Response: dear, you are correct that there are interesting frameworks to evaluate the implementation of a program, specifically the barriers and facilitators, including the Consolidated Framework for Implementation Research (CFIR) and Theoretical Domains Framework (TDF). Initially, we tried to design the tools and interview guide considering the constructs of CFIR, nevertheless, we decided that some of the constructs may not fit to our objective, and we believe that the access framework would fit our study. There is an updated framework developed by the WHO which supports the access framework (https://www.who.int/publications/i/item/9789241502894. Even the nomenclature of the framework and dimensions included are identical, ‘access framework’. We also decided that the qualitative part should be presented in using thematic analysis. Thank you for the wonderful insight, and we will think of applying such frameworks in our future work to better design tools/guide and report findings. 

6. Line 133 - the definition of 'availability' is confined to equipment/resources required to support the C-OTP. This seems at odds with the research aim and program provision. It is also not the definition held within the cited access framework reference. A reference is required for the Hawthorne effect (line 217).

Response: Dear, we can see your critical view. Yes, the definition of ‘availability’ is beyond checking the physical availability and functionality of essential medical equipment in a given healthcare institution. It would mean the service/program availability in a particular healthcare institution. As you said, this is clearly described in the cited article. This concept is partly addressed by our study’s ‘acceptability’ domain. That section addressed the waiting time, travel time, and so forth. Our study defines availability as the physical availability and functionality of essential equipment. For this, we used another framework developed by the World Health Organization (WHO) (https://www.who.int/publications/i/item/9789241502894). Still, the concept is under access framework. We have cited the article in the revised manuscript. 

As to the concern of providing reference to a strategy designed to mitigate the Hawthorne effect, we have modified the strategy that we used. It’s an interesting comment. We have been waiting for the opportunity to revise the statement based on the actual strategy we have applied during the actual data collection. Dear, the previous strategy was based on what we hear/learned from expertise and senior scholars, but we couldn’t get evidence to support it. Some scholars suggest that method (discarding recordings) if the design is ‘grounded theory’, which is not similar to our aim. We then change our minds and apply other strategies that are backed by evidence. The detail is presented in the revised version of the manuscript. 

7. What framework/theory will support the deductive analysis of the qualitative data? The authors mention 'existing literature' - has an evidence synthesis of existing literature been performed? (line 234-235).

Response: Dear, sorry for not updating the manuscript. Initially, as described earlier, we had planned to employ a framework, namely CFIR, to gather and analyse the data. However, after reviewing the constructs of this framework, we decided not to apply it. We rather planned to use thematic analysis and now we are analysing the data using thematic analysis referencing the Braun & Clarke’s guide for analysing qualitative data using the thematic analysis approach. https://www.tandfonline.com/doi/pdf/10.1191/1478088706qp063oa?needAccess=true

Yes, indeed, we have reviewed some literature to prepare our interview guide in order not to miss significant attributes/concepts that need to be explored. Although we didn’t review exactly similar studies, we have reviewed studies done in healthcare aiming at exploring factors affecting the level of implementation of health services/programs. Then experts in the field have tried to conceptualize the questions to prepare a flexible and comprehensive interview guide. Some themes will be generated based on the previous studies’ themes with or without modifications. For example, one of the elements of the interview guide we adopted from the literature was ‘miss use of therapeutic foods.’ While developing our guide, including this concept would help yield relevant information on the consumer’s side. Thus, we will have a theme stating the how mothers/caregivers use the given therapeutic food. 

8. What methodology does the 6-step process of qualitative analysis and coding refer to? (I'm assuming Braun & Clarke?).

Response: yes, we will follow Braun & Clarke’s six phases of qualitative thematic data analysis. https://www.tandfonline.com/doi/pdf/10.1191/1478088706qp063oa?needAccess=true. We have cited the article. 

9. Line 240: I am unclear whether the "questions and concerns" to discuss at the interview arise from previously completed surveys/questionnaires? A timeline to support the flow process of the evaluation study would help to clarify.

Response: Dear, you are correct that the message was confusing. It seems to be a sequential mixed-method approach will be used. We have modified the statement to avoid confusion. Dear, we have collected the data (both qual and quant) parallelly if possible. In exceptional cases, the quantitative preceded the qualitative data, but it doesn’t mean the approach is sequential instead, it’s to manage the data collection. 

10. This has much improved, but there are some unresolved grammatical errors. There are acronyms used throughout that are not fully explained at first use, e.g., MUAC. There is also inconsistency in the application of C-OTP (lines 6, 27 and 45). This should be double-checked throughout the manuscript. I would also suggest that appropriate language is considered in the discussion, line 273, and the use of "victims" to describe the population of interest in this research.

Response: comment accepted and a detailed revision has been made. 

Reviewer #2

1. Strengthen methods section of this protocol and needs clear explanation to understand project. The protocol has not discussed role and involvement of a statistician and the data analysis plans also vaguely discussed in this protocol (instead of ‘binary Binary logistic regression analysis will be conducted’, discuss/list out the type, definition and measurement of select outcomes/dependent variables along with possible independent variables).

Response: the comment is found to be valid and the necessary revision has been made. A significant change has been made.

2. Discuss the data collection and management process, data entry (electornic/double data entry, data quality validation and assurance

Response: Comment accepted and we have incorporated it in the revised document. The data quality assurance techniques are described in the ‘data collection tools, procedures, and quality assurance’ section. 

3. Once the data collection completed before the publication of the protocol, it is suggested to present the date of completion of data collection and final status of data along with any inclusions/exclusions, missing values, etc.

Response: Dear, your comment is appreciated. However, as you know in principle, a study protocol is supposed to be published/disseminated before the actual work begins. If we put the actual date of completion, excluded/included participants, and how the missing data was managed, we feel it seems a manuscript of an actual study and may confuse participants. Dear, all the plans regarding the above-mentioned points are already presented in the protocol. 

Scholars usually recommend considering an ‘amendment plan’ during preparing a protocol and some journals recommend to incorporate this although it is not available in any of the protocol reporting guides like PRISMA-P and SPIRIT. In this sense, to increase transparency, level of flexibility, and clarity, we have added an amendment section in the revised version, which is the recommended practice while preparing a protocol. However, if you feel the issue is major and needs to be considered, we are delighted to incorporate it before the protocol is published and released online. 

4. Discuss the filed level operation issues and lessons learnt; possible measures adopted etc., during the study data collection/implementation.

Response: Dear, we highly value your comment. We lack the confidence to incorporate the points raised here as we couldn’t find a reference to cite. It’s not associated with loss of trust. As you know, scholars are recommended to follow a scientific format to prepare and report studies. We try to review a protocol reporting guidelines; no guideline like SPRIT (for trail studies) and PRISMA-P (for systematic review and meta-analysis) that contains items stated regarding your points to be considered when preparing and disseminating a protocol once an actual study is finalized or begun. Nevertheless, if it’s your strong recommendation to incorporate this point in the protocol, we are ready to accept your comment. 

SPRIT: https://www.spirit-statement.org/wp-content/uploads/2013/01/SPIRIT-Checklist-download-8Jan13.pdf

PRISMA-P: https://pubmed.ncbi.nlm.nih.gov/25554246/

Note: these checklists are not relevant and applicable to our study. It’s just to show the concerns you raised are not common to be considered while preparing and publishing study protocols. 

Thank you!

Wubet Worku Takele (the corresponding author)

---

## [Decision Letter · Decision Letter 2]

26 Aug 2022

PONE-D-21-25442R2Evaluation of the community-based outpatient therapeutic feeding program implementation for managing children with severe acute malnutrition in Northwest Ethiopi a: a mixed-method evaluation protocol.

PLOS ONE

Dear Dr. Takele,

Thank you for submitting your manuscript to PLOS ONE. After careful consideration, we feel that it has merit but does not fully meet PLOS ONE’s publication criteria as it currently stands. Therefore, we invite you to submit a revised version of the manuscript that addresses the points raised during the review process.

The manuscript has been evaluated by two reviewers, and their comments are available below.

The reviewers have raised concerns regarding the reporting and methodology of this study.

Could you please revise the manuscript to carefully address the concerns raised?

We look forward to receiving your revised manuscript.

Kind regards,

Johannes Stortz

Staff Editor

PLOS ONE

Journal Requirements:

Reviewers' comments:

Reviewer's Responses to Questions

**Comments to the Author**

1. Does the manuscript provide a valid rationale for the proposed study, with clearly identified and justified research questions?

Reviewer #1: Yes

Reviewer #2: Yes

2. Is the protocol technically sound and planned in a manner that will lead to a meaningful outcome and allow testing the stated hypotheses?

Reviewer #1: Yes

Reviewer #2: Partly

3. Is the methodology feasible and described in sufficient detail to allow the work to be replicable?

Reviewer #1: Yes

Reviewer #2: No

4. Have the authors described where all data underlying the findings will be made available when the study is complete?

Reviewer #1: Yes

Reviewer #2: Yes

5. Is the manuscript presented in an intelligible fashion and written in standard English?

Reviewer #1: No

Reviewer #2: No

6. Review Comments to the Author

You may also provide optional suggestions and comments to authors that they might find helpful in planning their study.

Reviewer #1: This protocol aims to report the procedure and methods to evaluate a community-based programme delivering timely and therapeutic care to children who experience severe acute malnutrition. The authors seek to describe the barriers and enablers that impact on the implantation of a programme that integrates levels of health care and relies on the competence of health education workers working with families. Therefore, the protocol is required to describe how the complexity of programme implementation can be explored rigorously and contributes to open science. A strength of this paper is the authors use of mixed-methods towards a comprehensive and holistic exploration of the factors of programme implementation.

Thank you to the authors for their careful attention and further development of this manuscript. I have some minor issues with the manuscript that I outline in the uploaded word document that details suggested amendments to text.

Reviewer #2: The authors have made some revision to address the concerns raised in the first round of review process adequately. While the authors have made some amendments to the manuscript which improve the text, the concerns I have in the analysis of data /results, remain and failed to address the concerns raised in the first round of review process on integration/triangulation of results from both qualitative and quantitative approaches (mixed-methods), adequately. The authors suggested to discuss the rationale on using both qualitative and quantitative approaches and how they combine the elements of both qualitative and quantitative research approaches to address the study objectives. Since my primary concern is that the manuscript failed in triangulation/integration of results. Moreover, the manuscript contains insufficient data and explanations in addressing the crucial aspects of the study.

Among the minor concerns about the methodology, the authors suggested to discuss on the study setting, data collection process, selection and recruitment of sample participants etc., in order to avoid the data dredging. Moreover, I feel the manuscript is certainly less disjointed (study implementation, linkage with the qualitative results, utilization of results from both qualitative and quantitative data analysis) and I would urge the authors to have another good 'go' at the report, post-acceptance, as I am sure they will be able to make further minor improvements in structure and presentation. Also avoid typo errors (for instance, “Similarl,”; “Ethiopi a” etc.). I would also suggest someone who has statistical knowledge and, just to knock off the rough edges on the definition and measurement of the dependent variables like knowledge of HEWs about SAM; compliance of HEWs to the C-OTP protocol; acceptability of C-OTP of the service; severely malnourished children etc.,

In conclusion, the subject addressed in this study protocol manuscript is worth of investigation, nevertheless still more improvement could be sought and achieved via additional rounds of rigorous review; and acceptable after taking into account the above mentioned issues.

7. PLOS authors have the option to publish the peer review history of their article (what does this mean?). If published, this will include your full peer review and any attached files.

Reviewer #1: **Yes: **Dr Anne Griffin

Reviewer #2: **Yes: **Ramesh Poluru

---

## [Author Response · Author response to Decision Letter 2]

8 Sep 2022

Date: 08/09/2022

Point-by-point response 

Dear Editor and reviewers, thank you for giving us critical comments. The comments have a significant positive impact on improving the readability and acceptability of the manuscript. We have responded to each concern and question. An academic native speaker has edited the language. 

Kindly find it here below. 

Reviewer #1

1. Thank you to the authors for their careful attention and further development of this manuscript. I have some minor issues with the manuscript that I outline in the uploaded word document that details suggested amendments to the text.

Response: dear reviewer, we are grateful for the comments. We have fully incorporated the editorial and typo comments. 

2. Line 133-136: It is unclear if the focus group discussions will contribute to the qualitative data. Consider the position of the sentence about the logic of this section describing procedures.

Response: comment has been accepted, and the sentence has been re-arranged. 

3. Line 154, p6: clarification required that ‘good acceptance’ is defined as mothers answering to a consensus of “agree/strongly agree” rather than the total number of mothers responding to the questions (where any answer might be given). 

4. Response: Dear, if we get your concern, the definition needs to be based on the mother’s response, not the proportion of mothers/caregivers who have ‘agreed/ disagreed’. We have revised the statement to clarify more. 

5. Table 1: remove the gender pronouns from row 5 describing compliance/adherence measures.

Response: your comment has been accepted. 

6. Line 235: Please clarify if it is a nutritionist or a statistician who will perform the data analys.s? I assume that the nutritionist will be ably assisted by a statistician in analysing and interpreting the data based on the previously described methodology.

Response: Yes, you are correct that the nutritionist will do the analysis; we assume the nutritionist is knowledgeable about the variables and their definition; the nutritionist will do the analysis and be supported by a statistician. We have revised the document. 

7. Line 265-266: Can you define what is meant by “level the generated themes” – is it refining, defining and naming the themes?

Response: Dear, sorry for the typo error. It was to say ‘label/name’, not level. 

8. Line 11: suggest the use of the word ‘factors’ instead of “indicators” to describe those elements of the programme that can be evaluated for implementation and avoid the repetition in line 12.

. 

Response: Dear, we appreciate your perspective. Thank you for giving this comment. However, as can be recalled from our previous response, ‘indicator’ is an appropriate technical term used to describe the detail of programs. The evaluation is performed based on this key indicator. When we say ‘indicator’, we are not referring to factors(barriers/enablers) contributing to the program’s implementation. All in all, we prefer to use the term indicator over factor. Following your comment, we authors have discussed and have sought an explanation from one of the evaluation experts (co-author), and we have agreed to continue with ‘indicator’. 

Reviewer #2

1. While the authors have made some amendments to the manuscript which improve the text, the concerns I have in the analysis of data /results, remain and failed to address the concerns raised in the first round of the review process on integration/triangulation of results from both qualitative and quantitative approaches (mixed-methods), adequately. The authors suggested discussing the rationale for using both qualitative and quantitative approaches and how they combine the elements of both qualitative and quantitative research approaches to address the study objectives. Since my primary concern is that the manuscript failed in triangulation/integration of results. Moreover, the manuscript contains insufficient data and explanations in addressing the crucial aspects of the study.

Response: Dear reviewer, thank you for the comment. We believe we have provided rich information on how the qualitative and quantitative data will be integrated (see page 12). Dear, as you know, this is a protocol (it doesn’t include results), and we don’t have the data yet to show the actual integration. What we have discussed in this section is the plan. 

2. Among the minor concerns about the methodology, the authors suggested to discuss on the study setting, data collection process, selection and recruitment of sample participants etc., in order to avoid data dredging. Moreover, I feel the manuscript is certainly less disjointed (study implementation, linkage with the qualitative results, utilization of results from both qualitative and quantitative data analysis) and I would urge the authors to have another good 'go' at the report, post-acceptance, as I am sure they will be able to make further minor improvements in structure and presentation. Also avoid typo errors (for instance, “Similarl,”; “Ethiopi a” etc.). I would also suggest someone who has statistical knowledge and, just to knock off the rough edges on the definition and measurement of the dependent variables like knowledge of HEWs about SAM; compliance of HEWs to the C-OTP protocol; acceptability of C-OTP of the service; severely malnourished children etc.

Response: we have incorporated the editorial comments. For your information, we have proposed the measurement strategies after consulting a statistician. We will update the analysis if we find a revision is required after discussing it with co-authors, including the statistician. 

Thank you!

Wubet Worku Takele (the corresponding author).

---

## [Decision Letter · Decision Letter 3]

27 Sep 2022

Evaluation of the community-based outpatient therapeutic feeding program implementation for managing children with severe acute malnutrition in Northwest Ethiopi a: a mixed-method evaluation protocol.

PONE-D-21-25442R3

Dear Dr. Worku Takele,

We’re pleased to inform you that your manuscript has been judged scientifically suitable for publication and will be formally accepted for publication once it meets all outstanding technical requirements.

Kind regards,

Bassey E. Ebenso, Ph.D., M.P.H., M.D.,

Academic Editor

PLOS ONE

Additional Editor Comments (optional):

Your revised manuscript (Revision 3) has sufficently addressed reviewers comments.

Reviewers' comments:

Reviewer's Responses to Questions

**Comments to the Author**

1. Does the manuscript provide a valid rationale for the proposed study, with clearly identified and justified research questions?

Reviewer #2: Yes

2. Is the protocol technically sound and planned in a manner that will lead to a meaningful outcome and allow testing the stated hypotheses?

Reviewer #2: Yes

3. Is the methodology feasible and described in sufficient detail to allow the work to be replicable?

Reviewer #2: Yes

4. Have the authors described where all data underlying the findings will be made available when the study is complete?

Reviewer #2: No

5. Is the manuscript presented in an intelligible fashion and written in standard English?

Reviewer #2: Yes

6. Review Comments to the Author

You may also provide optional suggestions and comments to authors that they might find helpful in planning their study.

Reviewer #2: I would like to congratulate the author(s) to adequately addressing reviewers concerns to some extent satisfactorily. I would urge the authors to have another good 'go' at the manuscript, as I am sure they will be able to make further minor improvements in structure and presentation, just to knock off the rough edges.

In conclusion, the subject addressed in this protocol manuscript is worth of investigation and recommend for possible consideration.

7. PLOS authors have the option to publish the peer review history of their article (what does this mean?). If published, this will include your full peer review and any attached files.

Reviewer #2: **Yes: **Ramesh Poluru

---

## [Editor Report · Acceptance letter]

30 Sep 2022

PONE-D-21-25442R3 

Evaluation of the community-based outpatient therapeutic feeding program implementation for managing children with severe acute malnutrition in Northwest Ethiopia: a mixed-method evaluation protocol. 

Dear Dr. Takele:

I'm pleased to inform you that your manuscript has been deemed suitable for publication in PLOS ONE. Congratulations! Your manuscript is now with our production department. 

Kind regards, 

on behalf of

Dr. Bassey E. Ebenso 

Academic Editor

PLOS ONE